# Emission and Migration of Nanoscale Particles during Osseointegration and Disintegration of Dental Implants in the Clinic and Experiment and the Influence on Cytokine Production

**DOI:** 10.3390/ijms24119678

**Published:** 2023-06-02

**Authors:** Varvara Labis, Ernest Bazikyan, Svetlana Sizova, Vladimir Oleinikov, Andrey Trulioff, Maria Serebriakova, Igor Kudryavtsev, Dmitry Khmelenin, Olga Zhigalina, Irina Dyachkova, Denis Zolotov, Victor Asadchikov, Tatyana Mrugova, Aleksandr Zurochka, Sergey Khaidukov, Ivan G. Kozlov

**Affiliations:** 1Stomatology Faculty, A.I. Yevdokimov Moscow State University of Medicine and Dentistry, 20, p. 1 Delegatskaya St., 127473 Moscow, Russia; prof.bazikian@gmail.com; 2Shemyakin & Ovchinnikov Institute of Bioorganic Chemistry RAS, 16/10 Miklukho-Maklaya St., 117997 Moscow, Russia; sv.sizova@gmail.com (S.S.); voleinik@mail.ru (V.O.); khsergey54@mail.ru (S.K.); 3Federal State Budgetary Scientific Institution “Institute of Experimental Medicine”, 12 Acad. Pavlov St., 197022 Saint-Petersburg, Russia; trulioff@gmail.com (A.T.); m-serebryakova@yandex.ru (M.S.); igorek1981@yandex.ru (I.K.); 4National Medical Research Center of Oncology Named after N.N. Petrov of Ministry of Health of the Russian Federation, 68 Leningradskaya St., Pesochny, 197758 Saint-Petersburg, Russia; 5Federal Scientific Research Centre “Crystallography and Photonics” Russian Academy of Sciences, 59 Leninskiy Prospekt, 119333 Moscow, Russia; xorunn@gmail.com (D.K.); zhigal@crys.ras.ru (O.Z.); zolotovden@crys.ras.ru (D.Z.); asad@crys.ras.ru (V.A.); 6Department of Machine-Building Technologies, Bauman Moscow State Technical University, 5/1 2-ya Baumanskaya St., 105005 Moscow, Russia; 7Moscow State Clinical Hospital Named after V.V. Veresaev of the Moscow Healthcare Department, 10 Lobnenskaya St., 127644 Moscow, Russia; t.m.mrugova@gmail.com; 8Institute of Immunology and Physiology of the Ural Branch of the Russian Academy of Sciences, 106 Pervomaiskaya St., 620049 Ekaterinburg, Russia; av_zurochka@mail.ru; 9Laboratory of Immunobiotechnology of the Russian-Chinese Center for Systemic Pathology of SUSU (NRU), Federal State Autonomous Educational Institution of Higher Education “South Ural State University (National Research University)” 76, Lenin prospekt, 454080 Chelyabinsk, Russia; 10Institute of Professional Education, I.M. Sechenov First Moscow State Medical University, 8-2 Trubetskaya St., 119991 Moscow, Russia; immunopharmacology@yandex.ru

**Keywords:** nanoscale metal particles (NSMP), emission, migration, microbiota, modified basophil test, cytokines

## Abstract

The emission of nanoscale particles from the surfaces of dental implants leads to the cumulative effect of particle complexes in the bone bed and surrounding soft tissues. Aspects of particle migration with the possibility of their involvement in the development of pathological processes of systemic nature remain unexplored. The aim of this work was to study protein production during the interaction of immunocompetent cells with nanoscale metal particles obtained from the surfaces of dental implants in the supernatants. The ability to migrate nanoscale metal particles with possible involvement in the formation of pathological structures, in particular in the formation of gallstones, was also investigated. The following methods were used: microbiological studies, X-ray microtomography, X-ray fluorescence analysis, flow cytometry, electron microscopy, dynamic light scattering, and multiplex immunofluorescence analysis. For the first time, titanium nanoparticles in gallstones were identified by X-ray fluorescence analysis and electron microscopy with elemental mapping. The multiplex analysis method revealed that the physiological response of the immune system cells, in particular neutrophils, to nanosized metal particles significantly reduced TNF-a production both through direct interaction and through double lipopolysaccharide-induced signaling. For the first time, a significant decrease in TNF-a production was demonstrated when supernatants containing nanoscale metal particles were co-cultured with proinflammatory peritoneal exudate obtained from the peritoneum of the C57Bl/6J inbred mice line for one day.

## 1. Introduction

Despite the success of dental implantation as a method of treatment for patients with primary and secondary adentia in dentistry, it is necessary to identify a group of patients with complaints of unclear genesis. In the case of successful osseointegration of dental implants and functioning of orthopedic structures, in the delayed period, clinical manifestations occur in the form of xerostomia, burning of the oral mucosa, metallic taste in the mouth, and other symptoms. Patients attribute these manifestations to the placement of dental implants or orthopedic structures, requesting the unwarranted removal of dental implants or orthopedic structures, without performing the additional differential diagnostic tests required in such clinical cases. While doctors used to use medically certified products based on titanium alloy and were confident in the “bioinertness” of these materials [1], in the last decade there have been many scientific studies refuting this statement and pointing rather to immunoreactivity [2].

The authors’ studies [3] focus in particular on the ability to activate the NLRP3 inflammasome and synthesize IL-1β in response to nano- and microparticle titanium ions.

Studies have shown that the release of Ti particles/ions during implant insertion, early healing stages, late healing stages, and treatments during peri-implantitis might contribute to peri-implantitis through different mechanisms, such as foreign body reaction, cellular response, DNA methylation, and shaping the oral microbiome by increasing dysbiosis [4].

In studies [5], we studied the possibility of particles in the form of metal ions being emitted from the surfaces of dental implants when embedded directly into the bone bed on an animal model. The possibility of titanium ions emission at 1, 3, 7, 14, and 21 days was further studied. Cytocompatibility of these particles with fibroblast cultures HFF-1 and osteoblasts SAOS-2 was studied. The cytocompatibility studies showed that the particles are cytocompatible, but it is the smallest of them that have a lower and very close to 70% survival rate in both fibroblasts and osteoblasts.

The review [6] notes the fact that particle emission from the implant surface has been reported for the past decades. The role of the particles is still unexplored. Most often their size correlates with the range from 100 nm to 54 μm. The causes of particle emission are considered as mechanisms related to the placement of dental implants–friction, corrosion of the implant surface at the border with the abutment, and as a result of detoxification of the implant surface.

Nowadays, an increase in the oxide film on the surface of a functioning dental implant is associated with the toxicity of particles released as a result of biocorrosion, which correlates with the development of perimplantitis. It is necessary to pay attention to the presence of allergies before the placement of dental implants in patients [7].

Determination of critical parameters of transition from physiological processes during true osseointegration of dental implants to development of infectious and inflammatory complications in the form of mucositis and peri-implantitis was one of the tasks of our research group. We hypothesize that nanoscale metal particles (NSMP) are a major trigger for chronic immunopathological inflammation in the bone bed and oral mucosa tissues [8,9,10]. Their accumulation in tissues up to the critical mass we call “Critical Dose of NanoMetallic Particles” (CDNanoMP) leads to early death of own cells of the immune system, which is obviously the main trigger for accumulation of nanoparticles and formation of their complexes to micron size against the background of chronic immunopathological inflammation [11].

Among others, previous scientific studies [12] showed the possibility of migration of NSMP obtained in the supernatants from the surfaces of two systems of dental implants injected intraperitoneally in the experimental simulation of aseptic peritonitis in a mice model of the inbred *BALB/CJLaas* line. The ability to transport NSMP to immunocompetent organs, in particular to the spleen and their detection at the border of the red and white pulp, is an important aspect in understanding the processes of inactivation of intracellular antigens [13].

In order to understand the causes of complications in dental implantology, it is necessary to understand the physiological process of reparative osteogenesis from a nanotechnological and immunological perspective.

Studies aimed at the study of cytokine production as a result of intercellular interaction between NSMP and immunocompetent cells are a key link in the understanding of the mechanisms of osseointegration and disintegration of dental implants at the cellular-molecular level [14,15,16].

It is known that NSMP take part both in physiological and pathological reactions of the immune system in the process of reparative osteogenesis regulation.

Foreign authors study the problems of diagnosing possible allergic reactions to medical products based on titanium alloys [17,18,19,20,21,22]. It should be noted that a modified test for sensitisation or delayed-type hypersensitivity (DTH) to the surface of certified titanium alloy medical devices has now been developed under laboratory conditions before surgical intervention [23].

A multidisciplinary approach to diagnosing the causes of complaints of unclear genesis can rule out the unwarranted removal of dental implants. The study of the physiological response of the immune system to NSMP in experiment can help to understand the concept of normal osseointegration and pathological disintegration as reactions of the immune system to a foreign object in the form of a dental implant.

## 2. Results

### 2.1. Results of Clinical Case Investigation

#### 2.1.1. Results of X-ray and Electron Microscopic Studies

The result of the microtomography of the connective-tissue biopsy, above the cover screw of the Straumann dental implant placed in the position of the missing tooth 25 in patient Sh. (female, 52 years old), is shown in Figure 1. The scale shows the values of the linear absorption coefficient in mm^−1^.

As can be seen from the tomographic image of the connective-tissue biopsy, irregularly spaced foreign microinclusions of varying size and density are present in the area immediately adjacent to the implant.

The elemental composition of the detected microinclusions was determined by XRF. The results of the X-ray fluorescence measurements are shown in Figure 2. In all XRF spectra, the peaks from Ar (air), Cu, and Ag (X-ray tube anodes) are instrumental.

From the analysis of the obtained spectra, it can be concluded that Ti and Zr metal particles are present in the biopsy specimen, which is a consequence of their emission from functioning dental implants or the cover screw. Figure 3 and Figure 4 show the XRF spectra of the Straumann implant and the cover screw removed from the patient.

The spectra in Figure 3 and Figure 4 show that the main elements in the composition of the implant and the cover screw are titanium and zirconium.

In addition, the Straumann dental implant was examined using scanning electron microscopy with energy-dispersive analysis to clarify the elemental composition. Figure 5 shows SEM images and the corresponding energy-dispersive X-ray (EDX) spectrum obtained from the surface of the Straumann dental implant.

EDX spectra were obtained from different parts of the sample surface, one of which is shown in Figure 5c. The main elements of the spectrum are titanium and zirconium, which are determined in the amount of 85.44 and 10.47 weight percent; carbon is also present—4.08. Chemical elements such as calcium, oxygen, and chlorine may also be present in small quantities at different points on the surface. As a result, the elemental composition of the implant under study corresponds to the composition declared by the manufacturer, namely, the Roxolid material is a titanium-zirconium alloy consisting of ~15% Zr and ~85% Ti [24].

The conducted research proves the fact of NSMP emission into the gingiva of patient Sh. (female, 52 years old); however, it should be noted that as the fluorescence intensity is proportional to the fluorescent substance concentration, the concentration of Ti and Zr metal particles in the studied connective-tissue biopsy is small, which is proved by the low intensity of the mentioned elements peaks on fluorescence spectra (Figure 2).

To confirm the hypothesis of possible migration of metal nanoparticles in the human body and their accumulation in organs and tissues, the removed gallstones from patient Sh. (female, 52 years old) were examined using XMCT, XRF, and TEM methods, which formed during the use of osteointegrated dental implants as supports for the orthopedic structures. X-ray microtomography and fluorescence results of the gallstones are shown in Figure 6 and Figure 7. For the tomography images (Figure 6), the scale shows the linear absorption coefficient values in mm^−1^. XRF spectra (Figure 7) are presented for two of the stones and the other two are similar to the one presented.

The structure of gallstones is layered, with internal fissures and cavities. As can be seen from the microtomography images (Figure 6), gallstones contain inclusions with a higher absorption coefficient than in the volume. X-ray fluorescence measurements were carried out to detect Ti and Zr particles in the volume of gallstones under study. Fluorescence data (Figure 7) showed small amounts of titanium in all samples examined, but no clear traces of zirconium in the gallstones.

Figure 8 and Figure 9 show images of a single monocrystalline particle found in one of the gallstones, with a cut size of about 200 nm, covered by an amorphous shell 15–20 nm thick. Given the chemical element distribution maps (Figure 9b,c), the particle contains titanium, oxygen, and nitrogen, and the shell consists of carbon.

Figure 10a,b show images of a group of particles found in the gallstone containing particles of metals and their compounds: titanium oxide, a faceted particle with an elongated shape, measuring 400 × 200 nm, and particles containing iron, zinc, aluminium, silicon, and chlorine, measuring 50–150 nm. The corresponding EDX spectrum (Figure 10c) shows the elemental composition of this cluster and indicates the variety of chemical elements present there.

#### 2.1.2. Results of Oral Microbiological Examination of Patient Sh. (Female, 52 Years Old)

During the study, nine strains of microorganisms were isolated in the patient’s saliva, eight of which proved to be indigene species. When studying their species and quantitative characteristics (lg CFU/mL are given in brackets), it was found that the gram-positive cocci were represented by three strains from the genus *Streptococcus*: *Streptococcus salivarius* (5 lg CFU/mL), *Streptococcus vestibularis* (4 lg CFU/mL), and *Streptococcus oralis* (2 lg CFU/mL), and the gram-positive aerobic cocci *Rothia mucilaginosa* (3 lg CFU/mL). Gram-negative cocci were represented by the anaerobes *Veillonella parvula* (2 lg CFU/mL) and the aerobes *Neisseria subflava* (3 lg CFU/mL). In addition to the indigeneous cocci flora, strains of the gram-positive facultative anaerobic bacillus *Lactobacillus salivarius* (4 lg CFU/mL) and the obligate anaerobic gram-negative bacillus *Porphyromonas asaccharolytica* (3 lg CFU/mL) were isolated from saliva. No yeast-like fungi were detected in the saliva sample tested, nor were enterobacteria, non-fermenting Gram-negative bacteria, or staphylococci. However, an interesting fact was the isolation of a microorganism such as *Enterococcus cecorum* from the test material. The level of microbial infestation with this microorganism was 6 lg CFU/mL, which is significantly higher than the level of normal oral microflora isolated. This *Enterococcus* species is known to be representative of the normal intestinal microflora of chickens and a pathogen of various infections in poultry [25] and is also isolated from various animal species as a pathogen and commensal [26]. There is evidence in the literature of the ability of this organism to cause nosocomial infections in humans, and the number of such publications is increasing every year [27,28]. However, no data could be found on the isolation of *E. cecorum* from the oral cavity of humans, and in the case of patient Sh. (female, 52 years old), the role of this microorganism is also unclear. In antimicrobial sensitivity testing, the isolated strain was found to be sensitive to ampicillin, vancomycin, linezolid, tigecycline, rifampicin, and ciprofloxacin.

#### 2.1.3. Results of the Basophil Test of Patient Sh. (Female, 52 Years Old)

Figure 11 shows the results of the basophil test using histograms.

These histograms show the results of basophil activation at the time of examination of patient Sh. (female, 52 years old): Neg–control (initial level of activation of the patient’s venous blood basophils), Pos–positive test system control (activation of the patient’s venous blood basophils by co-culturing with the test system control), Straumann NSMP–co-culturing of the patient’s venous blood with a supernatant containing NSMP from the Straumann system dental implant surface; Astra Tech NSMP–co-culturing of the patient’s venous blood with the supernatant containing NSMP from the Astra Tech system dental implant surface; and Nobel Biocare NSMP–co-culturing of the patient’s venous blood with the supernatant containing NSMP from the Nobel Biocare system dental implant surface. According to the results of the basophil test, it can be noted the presence of sensitization to NSMP obtained in the supernatant from the surfaces of Nobel Biocare system dental implants. It should be noted that dental implants of the Straumann system implanted to the patient 3 years ago did not cause basophil degranulation at the NSMP level, which indicates that an allergic reaction to the surface of dental implants of this manufacturer is impossible. The activation index of basophils to the supernatant containing NSMP from Straumann dental implant surfaces is 0.86, relative to NSMP from Astra Tech dental implant surfaces is 0.72, and relative to NSMP from Nobel Biocare dental implant surfaces is 1.9.

The histograms in Figure 12 show the results of Th2 type activation, indicating the possibility of an allergic reaction in the patient, due to an increased content of this subpopulation of T-lymphocytes greater than 1.5%. Th2 type accounted for 2.22%, exceeding the maximum level by 0.72%. This result justifies the manifestation of food allergies noted by the patient in the form of atopic dermatitis, which manifested before the placement of dental implants.

### 2.2. Results of Cytokine Production Studies, When Co-Culturing NSMP with Inflammatory Peritoneal Cell Exudate Obtained in the Mice Model of C57Bl/6J Inbred Line

In the experiment, the parameters of nanoscale particles obtained in the supernatants from the surfaces of dental implants of the Nobel Replace system were determined by dynamic light scattering. The frequency of occurrence of particles in the supernatant is 9.2 kcps, the polydispersity coefficient is 0.378%, and the average size is 389.2 nm.

Studies of co-culture of Nobel Replace NSMP with mice peritonial exudate obtained on the first and third days, with and without lipopolysaccharide induction, were performed. The distribution by study groups is presented in Table 1.

There were a total of two study groups: 1 study group—with peritoneal exudate obtained on the first day when peritonitis was reproduced in a mice model; 2 study group—with peritoneal exudate obtained on the third day when peritonitis was reproduced in a mice model. There are a total of eight subgroups. Study group 1: four subgroups when cultured with one-day mice peritoneal exudate; Study group 2: four subgroups when cultured with three-day mice exudate. Each study group consisted of four subgroups: two control subgroups and two study subgroups. First control subgroup: peritoneal exudate; second control subgroup: peritoneal exudate with addition of Nobel Replace NSMP. The third study subgroup: peritoneal exudate when co-cultured with lipopolysaccharide; the fourth study subgroup: peritoneal exudate with Nobel Replace NSMP when co-cultured with lipopolysaccharide. The obtained results of cytokine intracellular cascade production by multiplex analysis using individual panels of test systems to determine IL-1B, IL-2, IL-4, IL-5, IL-6, IL-10, IFN-g, GM-CSF, IFN-a, RANTES, and TGF-b1 of the mice are shown in Figure 13 and Appendix A

Significant differences were revealed in the study groups in terms of intracellular TNF-a production during co-culture of Nobel Replace NSMP with proinflammatory mice exudate obtained on the first day and Nobel Replace NSMP induced by lipopolysaccharide compared to the control group, * *p* < 0.1, ** *p* < 0.01 (Figure 14).

From the obtained statistical results, it can be concluded that Nobel Replace NSMPs inhibit the intracellular production of TNF-a, respectively, have anti-inflammatory activity. Dual signaling induced by lipopolysaccharide induction reduces the anti-inflammatory effect of Nobel Replace NSMP.

Changes in other anti-inflammatory and proinflammatory cytokines in the study subgroups were found to be without significant significance when the data were statistically processed.

## 3. Discussion

For the presented clinical case, nano- and microscale particles impregnated in the connective-tissue graft taken over the cover screw and the Straumann system dental implant were identified; the structural and elemental composition of the dental implant as well as the cover screw removed from the patient’s oral cavity were studied. The elemental composition of the investigated implant of the Straumann system was found to correspond to the composition declared by the manufacturer. An interesting fact was the discovery of metal nanoparticles in the structure of the patient’s gallstones containing titanium, iron, and other microelements. Previous studies have investigated the mechanism of free emission of NSMP when reproducing the osseointegration of dental implants under laboratory conditions and revealed an increase in the oxide layer containing NSMP on the surfaces of dental implants when simulating physical loading [8]. The possibility of metal nanoparticles migration from the abdominal cavity into organs (in particular, the spleen) in a mice model of peritonitis has been shown [12,13], which, probably, due to nanoparticles get into the gastrointestinal tract, may contribute to the formation of crystallization foci for concrements formation. A microbiological study was conducted to verify pathogenic and opportunistic microflora in the oral cavity without detectable pathology. The study of the patient’s oral microbiota revealed changes characterized by a low level of contamination of mixed unstimulated saliva with microorganisms–representatives of the indigene microflora, as well as the isolation of *E. cecorum* strain–a microorganism, the isolation of which from people is extremely rare, and its role in the development of infectious processes is not studied. An allergic component to the surface of the removed Straumann dental implant was excluded as a cause of the patient’s complaints of unclear genesis.

In an experiment to study the effect of NSMP on the ability to induce cytokine production, when co-cultured with immunocompetent cells obtained from peritoneal exudate on a mice model of peritonitis, it was revealed that when co-cultured pro-inflammatory exudate, with a high content of neutrophils obtained on the first day, with NSMP, there was a significant decrease in TNF-a production. The results of this study revealed a decrease in TNF-a production in the subgroup when NSMP was co-cultured with neutrophils as opposed to controls, indicating an anti-inflammatory effect. This effect may be related both to the elemental composition of the particles themselves and to their frequency and size in the supernatant. In the subgroup of the study in which NSMP and LPS were co-cultured, the anti-inflammatory effect was comparable to the subgroup where the effect of NSMP on TNF-a production was studied. Presumably, NSMP have a pronounced anti-inflammatory effect. Thus, in the process of dental implant introduction and insignificant emission of NSMP included in the oxide layer, it can be assumed that there is a decrease in the ability of innate immunity cells to respond to a complex of antigens: bacteria (LPS) and NSMP.

## 4. Materials and Methods

### 4.1. Clinical Case

Patient Sh. (female), 52 years old, applied to the Department of Oral Surgery No 2 of Moscow State University of Medicine and Dentistry for complaints of unclear genesis in the form of xerostomia, burning in the mucous membrane of the tongue, and metallic taste in the mouth, particularly in the area of 4 units of dental implants Straumann system previously installed in a private clinic 3 years ago (in 2020). It should be noted that the patient has a history of chronic erosive ulcerative gastritis with gastroesophageal reflux, for which she was previously examined and treated by a gastroenterologist. Additionally, a chollicystectomy with removal of the gallbladder containing 4 stones was performed in 2022 (Figure 15).

One of the osseointegrated dental implants in tooth position 36 was removed at the patient’s insistence in a private clinic (Figure 16). Treatment with a gastroenterologist did not relieve the clinical symptoms, and the patient related his complaints to the previously installed and osseointegrated dental implants. Since these clinical symptoms appeared after dental implantation, the patient attributed them to the manifestation of an allergic reaction to dental implants, having a history of food allergies. This fact necessitated multidisciplinary research to confirm or rule out the association with dental implants as a cause of clinical manifestations in order to preserve them and use them as supports for orthopedic structures.

A connective-tissue graft was taken over the base of the cover screw in the projection of the dental implant placed in the place of the missing tooth 25 (Figure 17).

The structure of the removed dental implant was studied and its affiliation with the Straumann system was confirmed.

Patient’s Sh. (female, 52 years old) oral microbiota were analyzed on the basis of smears taken from the oral cavity in 5 points: mucosa of the cheeks, tongue (Figure 18), oropharynx, and alveolar process in the projection of dental implant 25.

A basophil test was performed to rule out sensitization to the alloy of a previously removed Straumann dental implant. Tissue biopsy material was studied to identify impregnated nanoscale and microscale particles in the structure of pathological tissues using X-ray microtomography and X-ray fluorescence analysis methods. By scanning electron microscopy with energy dispersive analysis, the surface composition of a remote medical certified product was identified. Using microbiological studies, the oral microbiota was studied, and candidiasis as a cause of clinical manifestations in the oral cavity was excluded. By flow cytometry using the “Allergenicity Kit, Cellular Analysis of Allergy” test system (Beckman Culture, Brea, CA, USA), the Straumann dental implant was excluded as the cause of the patient’s clinical symptoms.

### 4.2. X-ray Microtomography (XMCT)

X-ray studies of the connective-tissue biopsy sample located above the implant cover screw placed in the position of the missing tooth 25, as well as gallstones, were performed on a “TOMAS” microtomograph. [29]. Experiment parameters:Mo tube (focus size 12 × 0.4 mm), 40 kV × 40 mA mode;Monochromator—Pyrolytic graphite, reflection (0002);Wavelength—0.71 Å (E = 17.5 keV);Detector—Ximea XiRay11 (pixel size 9 × 9 microns, field of view 36 × 24 mm);Exposure—4 s per projection;Measuring range—800 projections in 0.25° increments (0°–200°).

### 4.3. X-ray Fluorescence Analysis (XRF)

The studies of the elemental composition of connective tissue biopsy sample located above the implant cover screw placed in the position of the missing tooth 25, cover screw, and gallstones were performed on X-ray microtomograph “DITOM-M” [30]. The anode of the tube was chosen depending on the elements to be analyzed. An X-ray tube with a copper anode was used to detect titanium particles (Ekα1 = 4.51 keV) in the objects under study (Experiment No. 1) and a tube with a silver anode was used for zirconium particles (Ekα1 = 15.775 keV) (Experiment No. 2).

Parameters of experiment No. 1:Cu tube (focus size 12×2.0 mm), 40 kV × 40 mA mode;Monochromator—silicon (symmetrical), reflection (111);Wavelength—1.54 Å (E = 8.047 keV);Beam size—10.0×1.0 mm (slits adjustable);Detector—Amptek 123SDD (Amptek, Bedford, MA, USA);Exposure—1200 s per measurement.

Parameters of experiment No. 2:Ag tube (focus size 10 × 1.0 mm), 40 kV × 40 mA mode;Monochromator—silicon (symmetrical), reflection (111);Wavelength—0.55 Å (E = 22.162 keV);Beam size—10.0 × 1.0 mm (slits adjustable);Detector—Amptek 123SDD (Amptek, Bedford, MA, USA);Exposure—1200 s per measurement.

### 4.4. Methods of Scanning Electron Microscopy (SEM) and Transmission Electron Microscopy (TEM) with Energy Dispersive (ED) Analysis

SEM studies were performed using a Scios Dual Beam scanning electron microscope (Thermofisher Scientific, Waltham, MA, USA) at an accelerating voltage of 20 kV in the secondary electron mode using an Everhart-Thornley detector. The samples were fixed in the holder with a carbon conductive tape.

TEM images were obtained using an Osiris transmission electron microscope (Thermofisher Scientific, Waltham, MA, USA) at an accelerating voltage of 200 kV. Gallstone samples were ground in a mortar to a powder, then dispersed in acetone in an ultrasonic bath, and applied to copper grids with a perforated carbon substrate.

### 4.5. Microbiological Study of the Oral Microbiota of Patient Sh. (Female, 52 Years Old)

To study the microbiota of the patient’s oral cavity, the study of the species and quantitative composition of the microflora of mixed unstimulated saliva, which is the integral environment of the oral cavity, was conducted. Sampling was performed on an empty stomach, observing the rules of asepsis before brushing teeth and treatment with antiseptic agents. Immediately before biomaterial sampling, the patient’s oral cavity was rinsed with 50 mL of physiological solution; the sampling was performed at rest with a preliminary accumulation of unstimulated mixed saliva for 2 min. Afterwards, saliva in a volume of 2 mL was collected by spitting into a sterile test tube (Nuova Aptaca, Canelli, Italy). The collected sample was delivered to the laboratory immediately. For the complex study of aerobic and anaerobic microflora, oral fluid was inoculated according to Gould’s method [31] on nutrient media of SIFIN diagnostics Gmb (Germany): Colombian agar with 5% lamb’s blood for isolation of streptococci and enterococci, yolk-salt agar for staphylococci, McConkey agar for enterobacteria, Saburo agar with dextrose for cultivation of yeast-like fungi, MRS agar for lactobacilli, and Shedler agar with 5% lamb’s blood for isolation of anaerobes. The crops were cultured under aerobic, anaerobic, and microaerophilic conditions under thermostatic conditions at 35–37 °C for 24–72 h using BD GasPak™ systems (BD, Franklin Lakes, NJ, USA) and appropriate gas generator bags, respectively. Species identification of isolated microorganisms was performed by matrix-activated laser desorption/ionization time-of-flight (MALDI-ToF) mass spectrometry using a MicroFlex LT mass spectrometer (MALDI Biotyper 3.1, Bruker Daltonik GmbH, Bremen, Germany). The identification at the species level with the manufacturer’s recommended limit value ≥2000 was considered reliable. The level of microbial infestation in saliva was determined by the number of colony-forming units (CFU) per 1 mL and expressed in logarithmic units (lg CFU/mL). Antimicrobial sensitivity was determined by the disk-diffusion method on Müller-Hinton agar (Bio-Rad, Marnes la Coquette, France) using standard disks (Bio-Rad, Marnes la Coquette, France).

### 4.6. Methodology for the Basophil Test

For this test, supernatants from the surfaces of three dental implant systems were obtained: Astra Tech, Straumann, and Nobel Replace, by the method described in the patent [23]. Supernatants containing NSMPs from the surface of TiO_2_-based dental implants Nobel Replace, Astra Tech were used as the controls and were obtained by the following method. The obtained supernatants containing nanosized metal particles fixed by dynamic light scattering were further used to perform a basophilic test.

In the work, the basophil test (“Allergenicity Kit, Cellular Analysis of Allergy”, Beckman Culture, Indianapolis, IN, USA) was performed to see the activation and degranulation of basophils by two molecules CD203, CD294 on the cell surface using monoclonal antibodies during joint culture of venous blood of the examined patients with the obtained nanosized metal particles as supernatants. Joint incubation of NSMPs and patient’s venous blood was performed for 15 min. Basophil activation or its absence was fixed by flow cytofluorimetry. Degeneration of basophils was detected by flow cytometry (flow cytometer FC-500). Based on the results obtained, the basophil activation index was calculated. In case of obtaining the ratio of the number of positive basophils of the studied sample to the number of negative basophils of the control sample ≤ 1, this implant is recommended for use.

Based on the percentage of CD203 expression on the cell membrane surface, one can judge the possibility, impossibility, and severity of an allergic component, including the degree of sensitization to the alloy used or the manufacturer’s certified TiO_2_-based metal product surface [23].

### 4.7. Experimental Laboratory Study

#### 4.7.1. Preparation of a Suspension of Nanoscale Particles from the Surface of Dental Implant System Nobel Replace

To obtain supernatants containing NSMP used in the experimental part of the study, 20 units of dental implants of systems Nobel Replace were incubated in bidistilled water for 5 days in a CO_2_ incubator. Nanoscale particles were obtained by the method described in the patent [23]. After incubation, tubes containing implants were treated with ultrasound at a frequency of 35 kHz for 20 min. Under the laminar conditions, the implants were removed from the tubes, the water was evaporated, and the remaining sediment containing nanoscale particles was diluted in phosphate-salt buffer solution (PBS), adjusting the concentration to 1/20 of the original volume. The resulting supernatant samples were filtered through a Millipore syringe nanofilter (maximum pore diameter D = 0.22 μm).

#### 4.7.2. Dynamic Light Scattering (DLS)

In the experiment, the parameters of nanoscale particles obtained in the supernatants from the surfaces of dental implants of the Nobel Replace system were determined by dynamic light scattering.

The method of DLS (photon correlation or quasi-elastic light scattering) is used to measure objects from 1 nm to 10 µm in size, which allows the detection of the yield of NSMP and MSMP (microscale metal particles) from objects in study [8].

To identify nanoscale particles from the surfaces of two dental implant systems in the obtained supernatants [32] and determine their size, a study was performed on 90 Plus Partical Size Analyzer (Brookhaven Instruments Corporation, Holtsville, NY, USA) in multimodal mode using automatic 90Plus/BI-MAS and “dust cut-off” functions that allows to subtract very large objects, particularly dust. The filter value was 20. The measurements were recorded at a temperature of 25 °C and a fixed light scattering angle of 661 nm laser.

Using the DLS method, three parameters were measured: NSMP diameter (D, nm), frequency of NSMP occurrence in the supernatants (ACR, kcps), and index of heterogeneity of NSMP distribution in supernatants—NSMP polydispersity (PD, %).

#### 4.7.3. Experiment on the Interaction of NSMP with Cells of Proinflammatory Exudate Obtained by Reproducing a Classical Model of Peritonitis in a Mice Model Using Peptone Broth

Using multiplex analysis, the effect of nanoscale metal particles on the ability to induce cytokine production, when co-cultured with immunocompetent cells derived from peritoneal exudate in a mice model of peritonitis, with the addition of inducers, was studied.

At the Carbohydrate Molecular Laboratory of the Shemyakin & Ovchinnikov Institute of Bioorganic Chemistry RAS (Moscow, Russia), from 20 dental implants of the Nobel Replace system, supernatants containing NSMP were obtained by the developed and previously described method in bidiistillate, at the rate of 1 mL bidiistillate per 1 test tube containing a 5.0–10 mm dental implant, thereby reducing the solvent volume by half. Subsequently, the supernatants obtained in 20 test tubes were combined into a single glass flask. It was evaporated and 2 mL of phosphate-buffered saline (PBS) was added, reducing the particle concentration by a factor of 10 and transferring the supernatant from the bidistillate to PBS. Pouring the supernatant into sterile tubes and sealing them with cotton-gauze swabs, it was sterilized in an autoclave without vacuum aspiration.

At the next stage of the experiment, 2 weeks after transporting forty units of *C57Bl/6J* inbred line mice to the vivarium of the Immunology Department of FSBSI “Institute of Experimental Medicine” (St. Petersburg, Russia), the experiment on the animal model was initiated. Inflammatory peritoneal cell exudate of mice was used for the experiment. Peritoneal inflammation was initiated by intraperitoneal injection of 2 mL of sterile peptone water. Peptone water was prepared according to the method given in [33], namely, 5 g of chemically pure table salt and 10 g of peptone dry matter were dissolved in 1 L of bidistilled water. The resulting solution was stirred thoroughly and sterilized at an overpressure of 0.1 MPa for 10 min.

Animals were killed by ether anesthesia to obtain peritoneal exudate. Next, 2 mL (400 units of cold PBS per 80 mL of heparin) were injected into the abdominal cavity and the resulting cell suspension was taken with syringes and transferred to polypropylene tubes. In half of the animals (*n* = 20), peritoneal exudate was taken 24 h after peptone water injection, in the remaining animals (*n* = 20) after 3 days. This separation was due to obtaining different cellular composition of the peritoneal exudate, as the number of neutrophils decreases, and the number of macrophages increases at a later date.

From each sample, 4 million cells were taken, perfused with RPMI 1640 medium to an equal volume, and then the cells were centrifuged for 5 min at 200 g. The supernatant was removed, and the cell precipitate was transferred into 4 mL of RPMI-1640 culture medium (Biolot, St. Petersburg, Russia) supplemented with 10% fetal bovine serum (FBS; Gibco, Thermo Fisher Scientific Inc., Bartlesville, OK, USA), 50 µg/mL gentamicin (Biolot, Russia), and 2 mM L-glutamine (Biolot, Russia).

The resulting cell suspension was added to the wells of a 96-well plate. Four comparison groups were formed from each sample with threefold replication: (1) intact proinflammatory exudate; (2) proinflammatory exudate with the addition of nanoscale particles; (3) proinflammatory exudate with the addition of lipopolysaccharide (LPS) from Escherichia coli (Sigma-Aldrich, Merck KGaA, Darmstadt, Germany); and (4) proinflammatory exudate induced by lipopolysaccharide with the addition of nanoscale particles.

For ascites samples obtained 24 h after induction of inflammation, exudates were cultured under standard conditions (37 °C, 5% CO_2_) for 6 h. At the end of incubation, the contents of the wells were transferred to microtubes and centrifuged for 10 min at 200 g. The supernatants were transferred to other microtubes and frozen at −70 °C.

For ascites samples obtained three days after induction of inflammation, exudates were cultured under standard conditions (37 °C, 5% CO_2_) for 24 h. Other manipulations were performed similar to those described above.

Subsequently, tubes with sera were transported in a portable refrigerator to the joint laboratory of immunobiotechnology of the Russian-Chinese Center of Systemic Pathology at “South Ural State University (National Research University)” (Chelyabinsk, Russia) and the laboratory of inflammatory immunology at the Institute of Immunology and Physiology of the Ural Branch of Russian Academy of Sciences (Ekaterinburg, Russia). Sera were tested for cytokine content using Bio-Rad test systems (Hercules, CA, USA) and MAGPIX-100 Bio-Rad instrument (USA) using multiplex immunofluorescence analysis (MIFA) method.

## 5. Conclusions

In the case study, using a personalised, multidisciplinary approach to examine the causal relationship to the occurrence of symptoms of unclear genesis in the patient after multiple dental implants, the removal of osseointegrated dental implants was excluded. The dental implants were not proven to be the cause of the patient’s symptoms of unclear genesis. Despite this fact, it should be noted that emission and migration of nanoscale metal particles corresponding to the composition of the dental implant and cover screw surfaces into the gingival biopsy material with an accumulative effect at the contact site were identified. Including titanium nanoparticles were found in the gallstone structure of this patient two years after her dental implants. It is important to pay attention to the gastrointestinal tract disease, in particular chronic gastritis with increased acidity in the anamnesis of patient Sh. (female, 52 years old). In our opinion, patient’s complaints substantiate the presence of gastrointestinal somatic pathology complicated by persistence of Enterococcus cecorum which aggravates mucosal immunity reactions due to immunosuppressive properties of this microorganism. Removal of the dental implant was not feasible.

An experimental study demonstrated for the first time a significant reduction in TNF-a production when supernatants containing nanosized metal particles derived from the surface of Nobel Replace dental implants were co-cultured with pro-inflammatory peritoneal exudate isolated from the abdomen of inbred mice on day 1 of the experiment, and when co-culturing NSMP with LPS. It is important to note the competitive anti-inflammatory effect of NSMP co-cultured with LPS in the third subgroup of the study in contrast to the control group and the fourth subgroup, where only neutrophils were co-cultured with LPS. The detection of an anti-inflammatory effect of nanosized metal particles, which are contained in small amounts, may interpret the positive success of immediate dental implantation. Presumably, not only the elemental composition and size, but also the amount of particles in the tissue may be a key aspect of the switching physiological responses of the immune system in response to nanoscale metal particles.

## Figures and Tables

**Figure 1 ijms-24-09678-f001:**
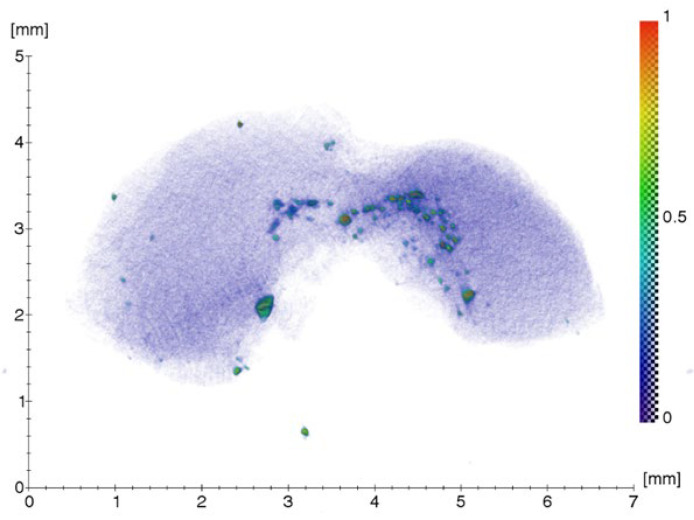
Result of the 3D reconstruction of the connective-tissue biopsy, above the implant cover screw, placed at the position of the missing tooth 25.

**Figure 2 ijms-24-09678-f002:**
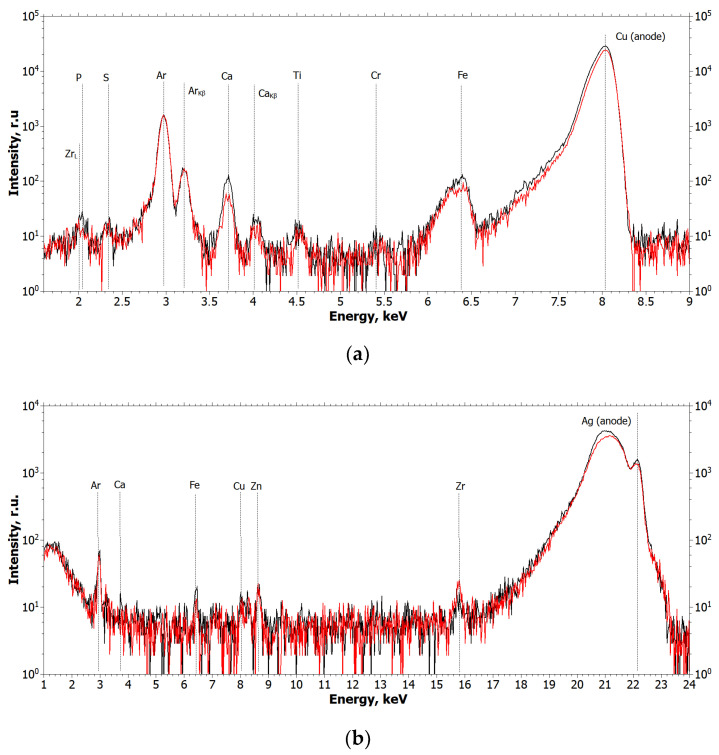
Fluorescence spectra of the biopsy sample at different points (black and red curves): (**a**) the wavelength is 1.54 Å (E = 8.047 keV); (**b**) the wavelength is 0.55 Å (E = 22.162 keV).

**Figure 3 ijms-24-09678-f003:**
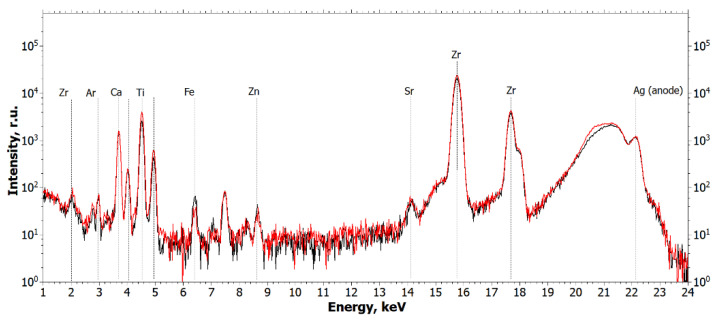
Fluorescence spectra of the Straumann implant measured at two different points (black and red curves). The wavelength is 0.55 Å (E = 22.162 keV).

**Figure 4 ijms-24-09678-f004:**
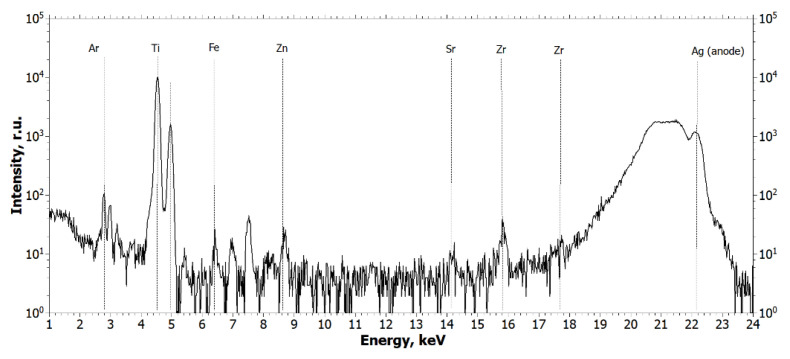
Fluorescence spectra of the cover screw. The wavelength is 0.55 Å (E = 22.162 keV).

**Figure 5 ijms-24-09678-f005:**
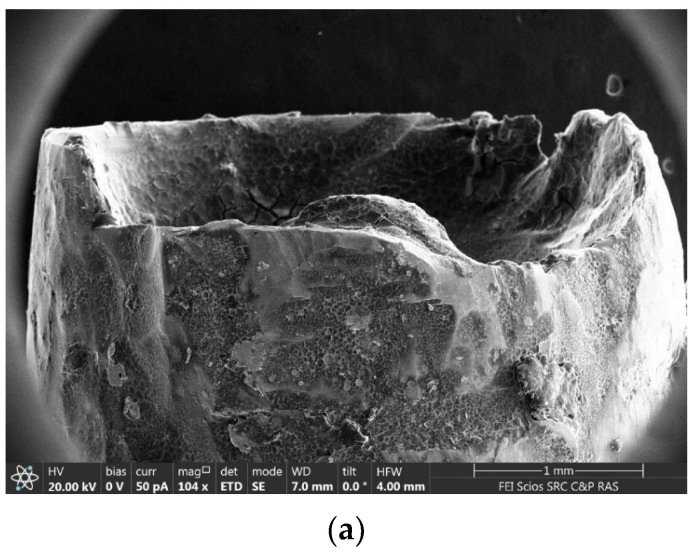
SEM images of the upper part of the implant: (**a**) a general view; (**b**,**c**) an enlarged image of the place where the EDX spectrum was taken from.

**Figure 6 ijms-24-09678-f006:**
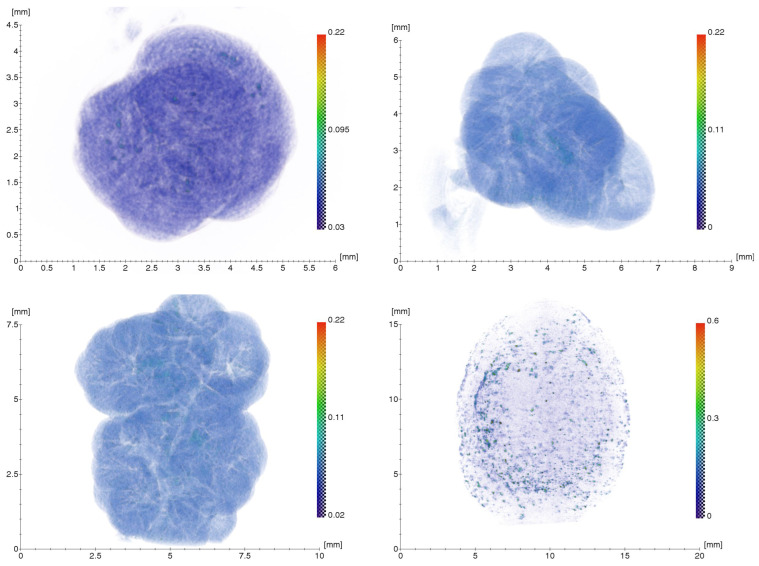
Three-dimensional reconstruction results for four gallstone samples.

**Figure 7 ijms-24-09678-f007:**
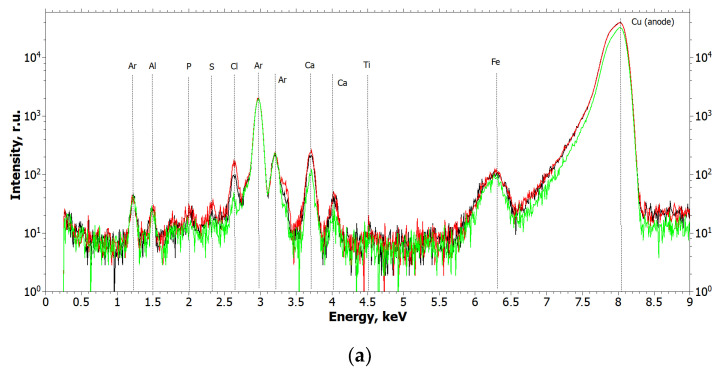
Fluorescence spectra of gallstone samples: (**a**) curves measured at different points of the sample (green and red curves). The wavelength is 1.54 Å (E = 8.047 keV); (**b**) the wavelength is 0.55 Å (E = 22.162 keV).

**Figure 8 ijms-24-09678-f008:**
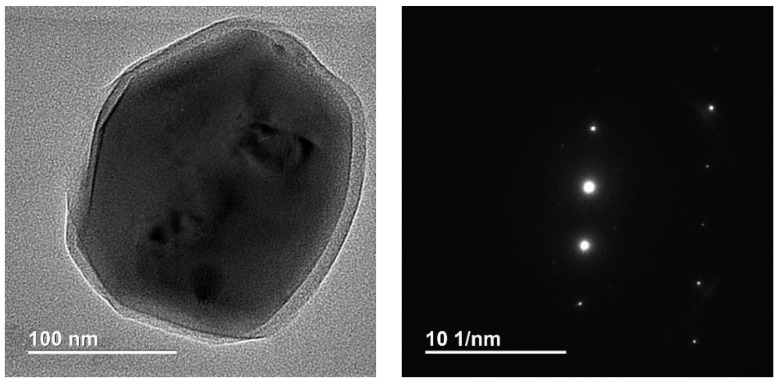
TEM image of a titanium-containing particle and the corresponding microelectronogram.

**Figure 9 ijms-24-09678-f009:**
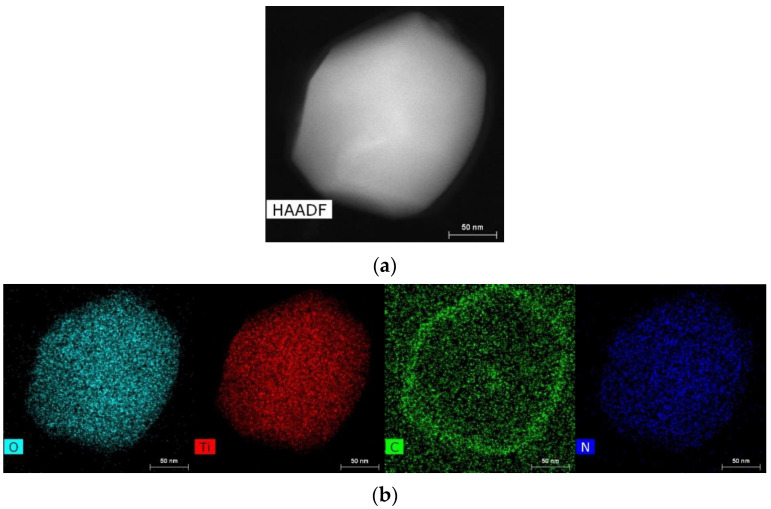
HAADF STEM image (**a**), chemical element distribution maps (**b**), and EDX spectrum (**c**) obtained from the particle in Figure 8.

**Figure 10 ijms-24-09678-f010:**
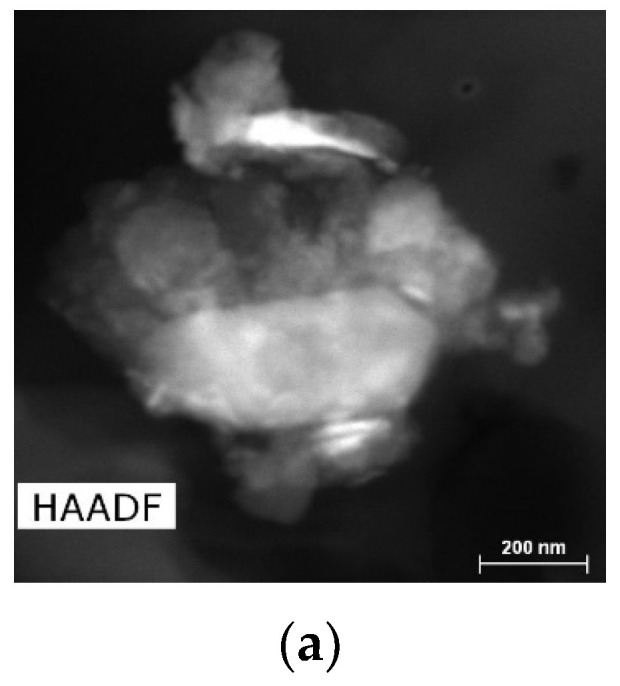
HAADF STEM image (**a**), chemical element distribution maps (**b**), and EDX spectrum (**c**) obtained from the particle cluster.

**Figure 11 ijms-24-09678-f011:**
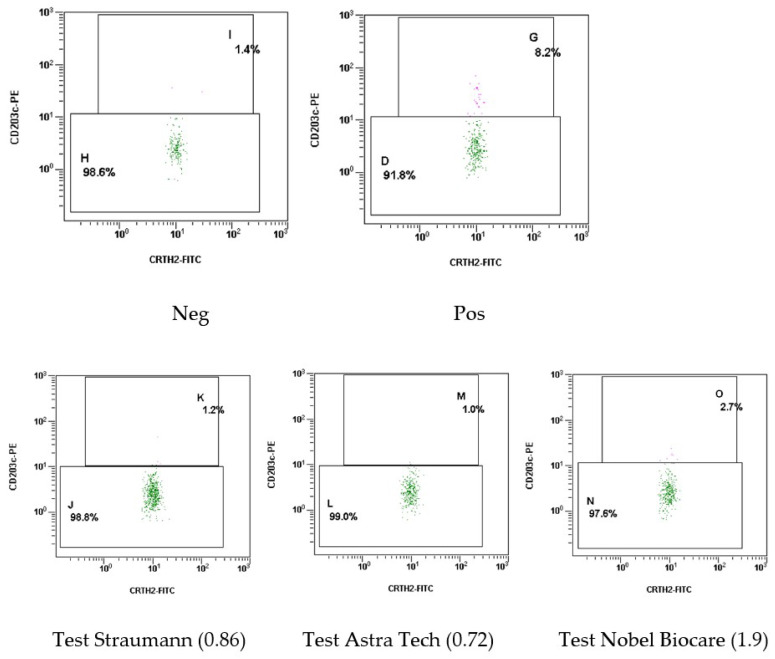
Histograms with the results of activation of basophils in the venous blood of patient Sh. (female, 52 years old) during a modified basophil allergy test.

**Figure 12 ijms-24-09678-f012:**
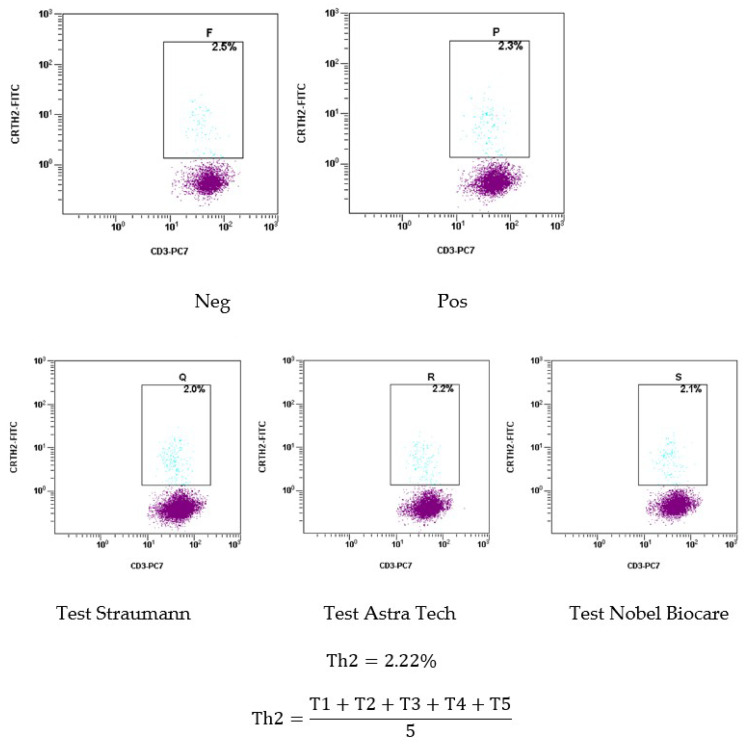
Histograms with the results of Th2 type activation.

**Figure 13 ijms-24-09678-f013:**
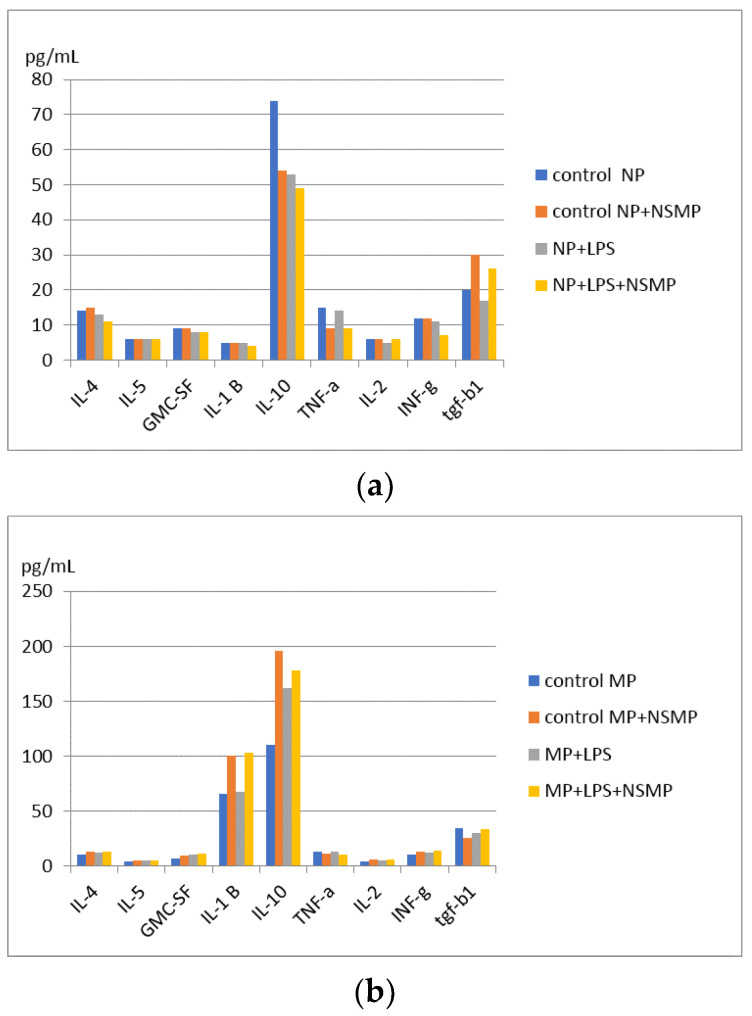
Changes in production of IL-4, IL-5, GMC-SF, IL-1B, IL-10, TNF-a, IL-2, INF-g, and TGF-b1: (**a**) on the first day in the four study subgroups; (**b**) on the third day in the four study subgroups.

**Figure 14 ijms-24-09678-f014:**
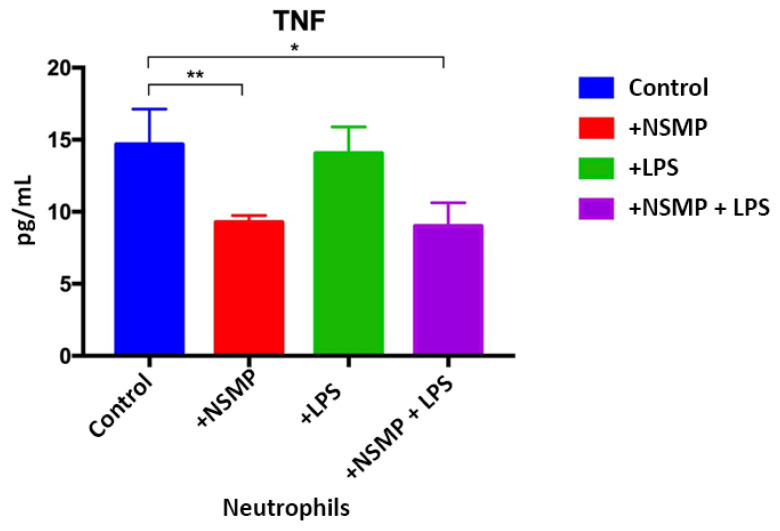
Distribution of TNF-a production data in the four subgroups during co-culture of Nobel Replace NSMP with proinflammatory exudate obtained on the first day. * *p* < 0.1, ** *p* < 0.01.

**Figure 15 ijms-24-09678-f015:**
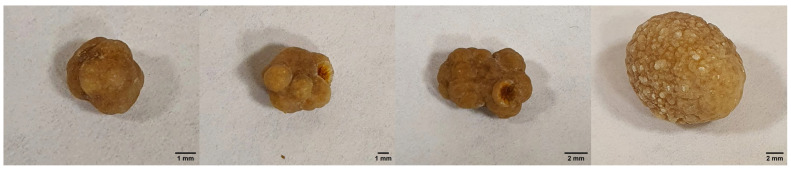
Photos of gallstone samples.

**Figure 16 ijms-24-09678-f016:**
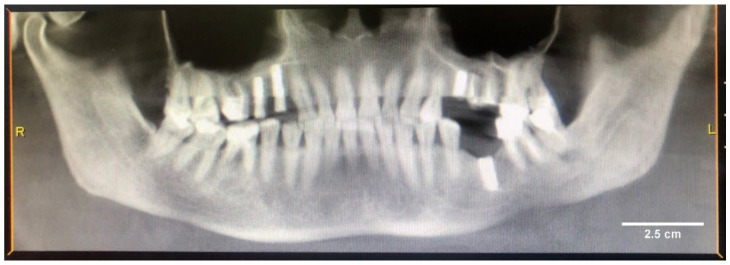
Orthopantomogram of patient Sh. (female, 52 years old) before removal of the dental implant in the position of the missing tooth 36.

**Figure 17 ijms-24-09678-f017:**
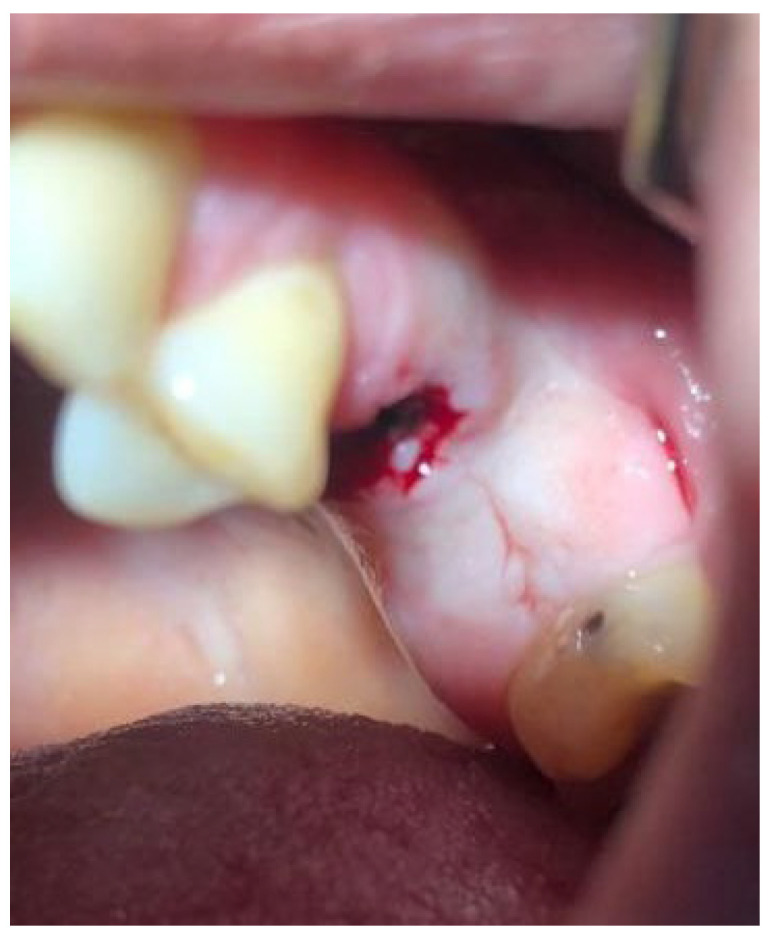
The connective tissue biopsy of patient Sh. (female, 52 years old) in the projection of the missing tooth 25 over the surface of the subsequently installed in its place osseointegrated dental implant of the Straumann system with a cover screw.

**Figure 18 ijms-24-09678-f018:**
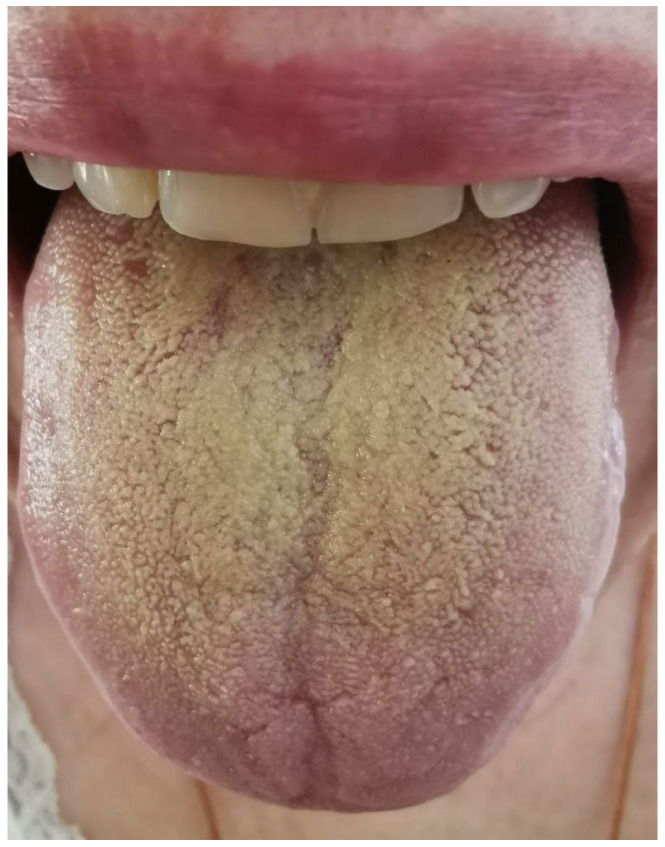
The condition of the mucous membrane of the tongue of patient Sh. (female, 52 years old) at the time of clinical examination.

**Table 1 ijms-24-09678-t001:** Distribution of study groups and controls when performing co-cultivation of NSMP obtained in the supernatants from the surfaces of Nobel Replace dental implants with peritoneal exudate isolated on the first and third days in the experimental modeling of peritonitis on the mice model of the *C57Bl/6J* inbred line.

Group 1 Peritoneal Cellular Exudate Obtained on the First Day				Group 2 Peritoneal Cellular Exudate Obtained on the Third Day			
Control 1	Control 2	Study 3	Study 4	Control 5	Control 6	Study 7	Study 8
Neutrophils (NP)	NP+ NSMP	NP+ LPS	NP+ LPS+ NSMP	Macrophages (MP)	MP+ NSMP	MP+ LPS	MP+ LPS+ NSMP

## Data Availability

The data presented in this study are available on request from the corresponding author. The data are not publicly available because they are part of the patent application, they cannot be provided to a wide range of the professional community until it is received by the authors. In particular, the research results are part of the dissertation work that has not been defended. After the defense of the scientific qualification work, the data can be published in the public domain.

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
