# Peer review of "Emission and Migration of Nanoscale Particles during Osseointegration and Disintegration of Dental Implants in the Clinic and Experiment and the Influence on Cytokine Production"

_ijms, 2023, doi:10.3390/ijms24119678_

Round 1

Reviewer 1 Report

Reviewer’s comments:

1.      This paper was interesting in the field of dental implantology, but needs major revisions so that readers can better understand authors’ new findings and ideas.

2.      Abstract should be re-written and re-organized. It should be stated that nano metal particles (NSMP) existed near dental implants. The particles were also transferred to gallstones.

3.      It is better to clearly distinguish experimental approaches to 2 major parts. 1) Finding of nano metallic particles (NSMP) near implants. Finding of these particles in gallstone of a patient. Microbial diagnosis of the patient having gallstones. 2) Cell culture tests of mouse-derived cells subject to NSMP and LPS.

4.      It is necessary to more explain methods of cell culture tests. The composition of culture medium, and the strain of LPS, etc.

5.      If possible, please add standard deviations of secreted cytokine amounts in Figure 13.

6.      The production method of NSMP was mentioned in the referred patent [23] and vague. It is hoped to describe the protocol in details.

7.      Multiplex cytokine studies were meaningful. Could you add gene expression studies?

8.      In discussion, please mention how NSMP was produced in the patient mouth having the implant, and the process how these particles were transferred to gallstone.

9.      In discussion, please think the mechanism why NSMP and LPS reduced TNF-a productionsof mouse-derived cells (neutophils). Did the decline change signaling magnitudes of each genes on the Toll-like (receptor) signaling pathway? Also, what is the clinical meaning caused by this decline?

English was OK. It is, howevr, highly expected to consider the underlying machanism and process related to  their findings. Consideration was absolutely lacked.

Reviewer 2 Report

1. The title of the study is not indicative of its content and there is no mention of osseointegration and disintegration of dental implants in the title.

2. Keywords should be corrected, keywords do not need a summary name. There are too many keywords.

3. In line 62, there are bulk references that should be spread between their respective paragraphs.

4. Lines 79 and 82 have no references.

5. The conclusion is not written for the abstract.

6. What does "patient Sh" mean?
7. The use of patients with gallstones should be written in the abstract.
8. Replace the material and method with the results because it causes confusion.
9. The following references may be useful:
 DOI: 10.1016/j.biopha.2018.09.026
DOI: 10.3390/ijms24032267
DOI: 10.1080/15376516.2019.1566424

Minor editing of English language required

Round 2

Reviewer 1 Report

The mamuscript was well improved and acceptable, but is still difficult for readers to entirely undestand. Please consider two following points. 1) In Abstract, please re-write the sentences so that readers can  comprehend the experimental approach and authors' findings in easier and more plain forms. 2) In conclusion, please re-organize the sentences. The consideration and speculation should ne transferred to Discussion section. Please state the facts (experimental findings) only in Conclusion.

It is hoped that DMPI check English of the manuscript prior to publication. 

Reviewer 2 Report

All corrections have been made.

All corrections have been made.